# Using Unconventional Wisdom to Re-Assess and Rebuild the BRICS

**Bertrand Guillotin**

Fox School of Business, Temple University, Philadelphia, PA 19122, USA; bertrand@temple.edu;
Tel.: +1-215-204-4201

**Abstract:** In 2015, Goldman Sachs closed its BRIC (Brazil, Russia, India, China) fund after years of losses and plummeting assets. Emerging markets had, once again, turned into submerging markets. Their dependence on "developed" markets and established institutions had failed them in a post-Global Financial Crisis (GFC) era, anchored in protectionism, risks, volatility, and uncertainty. The once commonly-accepted wisdom that called for US housing prices to always increase was part of the problem and contagion. Rebuilding the BRICS (S for South Africa) using conventional wisdom would probably not work. A new approach is necessary, especially since the last key contributions to show the inadequacy of a conventional wisdom-based strategy in emerging markets are more than ten years old. To help fill this gap, this paper proposes a holistic analytical framework for strategists to re-assess risks and opportunities in the BRICS. We illustrate how five basic assumptions can be proven wrong and lead to the creation of unconventional wisdom that can help derive some strategic insights. We find that rebuilding the BRICS for them to be more resilient is possible, if not vital, for the health of the global economy.

**Keywords:** emerging markets; globalization; protectionism

## 1. Introduction

In 2001, an economist at Goldman Sachs by the name of Jim O'Neill coined the BRIC term based on a paper that would change the world of investing and how we perceive risks in emerging markets (O'Neill 2001). Through his analysis, O'Neill, now Baron O'Neill of Gatley, convinced millions of investors that the opportunities outweighed the risks. The analysis was so timely that it became a type of conventional wisdom, expected to "show the path" for emerging markets through 2050 (Wilson and Purushothaman 2003). However, the path came to a dead-end in 2015 when Goldman Sachs closed the BRIC fund after years of losses and plummeting assets[1], about a year after O'Neill's retirement.[2] Whereas other acronyms have been invented (e.g., Next Eleven), none of them have managed to be as convincing and widely used.

In 2001 as well, Khanna and Rivkin's quantitative study of 12 emerging markets (3 out 5 BRICS were represented) confirms the inadequacy of conventional wisdom in advanced economies that "unrelated diversification depresses profitability" (Khanna and Rivkin 2001, p. 45). More specifically, these Harvard Business School scholars demonstrate statistically that "business group[3] affiliates earn higher accounting profits than do otherwise comparable unaffiliated firms" in several of the markets

---

1   https://www.ft.com/content/89f59acc-8679-11e5-8a12-b0ce506400af.
2   https://www.reuters.com/article/us-goldman-oneill/goldman-sachs-oneill-aka-mr-bric-to-retire-idUSBRE91411320130205.
3   Definition: "a business group is a set of firms which, though legally independent, are bound together by a constellation of formal and informal ties and are accustomed to taking coordinated action" (Khanna and Rivkin 2001, pp. 47–48).

examined (Khanna and Rivkin 2001, p. 68). This important study, published in the *Strategic Management Journal*, led to more contributions on emerging markets and was followed by two instrumental studies (Wright et al. 2005; London and Hart 2004).

In "Strategy Research in Emerging Economies: Challenging the Conventional Wisdom," Wright et al. point out the rising prominence of emerging economies in the world economy and explain the limitations of theoretical contributions based on institutional theory alone. By combining four strategies[4] in emerging economies with four theoretical perspectives[5], they give strategists a multidimensional lens for them to better understand emerging markets while challenging conventional wisdom. Their study builds on London and Hart (2004), whose exploratory analysis based on interviews of MNC managers, 24 original case studies, and archival materials, challenges the transnational model by suggesting that "Western-style patterns of economic development may not occur in low-income [emerging] markets" (London and Hart 2004, p. 350). Furthermore, they find that that "successful strategies suggest the importance of MNCs developing a global capacity in social embeddedness" through relationships with non-traditional partners (London and Hart 2004, p. 350).

Since 2005, no major contributions have been published to show strategists how to navigate a new normal where conventional wisdom no longer holds true. In our new normal, uncertainty, volatility, protectionism, and risks abound in the aftermath of the Brexit vote and the surprise election of Donald Trump that created the BRUMP effect (Ghemawat 2018). Therefore, a closer look at practice and a combination of theory and practice might be a source of greater insights.

In fact, whereas the 2008 GFC erased trillions of dollars in wealth, John Paulson made billions with his bet against the conventional wisdom on subprime loans and housing (Zuckerman 2010). The GFC, not only taught just about everyone the limitations of that wisdom, but also put institutions in the hot seat for having failed to protect investors and for being associated with the root cause of the main issue: greed (Pandit 2018). Specifically, one of the oldest institutions in the world received much criticism: the university and its business school (Giacalone and Wargo 2009; Howard and Cornuel 2012; Currie et al. 2010), the main source of talent on Wall Street. Therefore, in the post GFC era, it might be expected to marginalize business schools after this devastating crisis, in the same way that a comeback to protectionism—supposed to make things better at home—appears to be the most appropriate reaction against globalization and its risks. We think otherwise and challenge conventional wisdom on emerging and "developed" markets by using a holistic framework and five typical assumptions as our basis for illustration. We then carry out a comparative analysis of events and patterns across countries, institutions, and industries, in order to see if those assumptions are substantiated or invalidated in a broader and more complex context between 2000 and 2018, basing our inductive approach on a few theoretical framework and scholarly contributions.

This paper provides a simple framework to generate strategic insights to strategists and leaders who seek to re-assess the risks and opportunities in emerging markets and increase their resilience, essentially rebuilding the BRICS. Using the resulting unconventional wisdom, this rebuilding is, not only vital for the economic health of these emerging countries, but also for that of the global economy.

## 2. Materials and Methods

In this section, we discuss our inductive research approach and seek to answer how to derive any strategic insights going beyond the often-misleading conventional wisdom. Five assumptions associated with conventional wisdom have been formulating, as illustration, by the author who relies on his research and 25 years of international experience, including 10 years in corporate positions and

---

[4]　(1) Firms from developed economies entering emerging economies, (2) domestic firms competing within emerging economies, (3) firms from emerging economies entering other emerging economies, (4) firms from emerging economies entering developed economies" (Wright et al. 2005, p. 1).

[5]　Institutional theory, transaction cost theory, resource-based theory, and agency theory (Wright et al. 2005, p. 1).

15 years in academia.[6] It must be noted that these assumptions serve as examples to demonstrate the dangers of conventional wisdom. Each strategist must go deeper in the analysis, using the suggested framework below (please see Figure 1), as well as tools provided by well-respected scholars, such as Michael E. Porter, Pankaj Ghemawat, and others cited in this paper and in the literature.

In order to start our approach in the conventional wisdom realm, we must define what that is. According to the respected Merriam-Webster dictionary, conventional wisdom is "the generally accepted belief, opinion, judgment, or prediction about a particular matter (example: Conventional wisdom in Hollywood says that a movie can't succeed unless it stars a famous actor or actress)."[7] The basis for each assumption will be strengthened by at least one relevant academic reference.

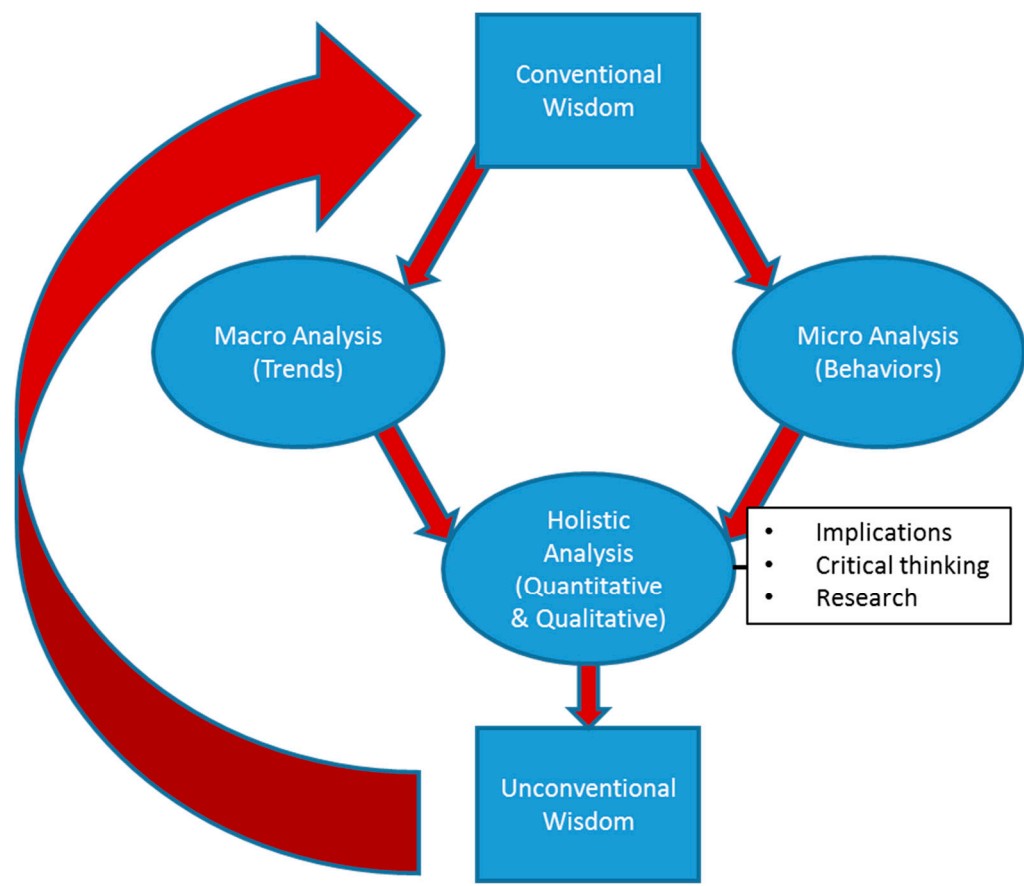

**Figure 1.** Holistic Risk and Opportunity Analytical Framework (HROAF).

**(A) Five assumptions and conventional wisdom**

**Assumption 1.** *In an increasingly protectionist world, a domestic mindset is the way to go.*

According to some respected scholars, globalization died after the GFC (Rugman 2012). As a result and since 2016, both the UK (Brexit vote) and the US (America First foreign policy) have implemented domestically-oriented policies to protect domestic jobs, especially by imposing tariffs on steel and

---

aluminum for the latter (Galbraith 2018). Other protectionist measures have included anti-immigration rhetoric and a re-focus on nationalism and patriotism in these countries and in others (e.g., Hungary, Italy, and Austria). In essence, globalization, once the notion that called for integration and collaboration in business and lifted hundreds of millions of people out of poverty (Ahlstrom 2010), has now become defunct for some and the enemy for others, even if the latter do not fully grasp the benefits that they have been deriving from it (Betts 2016).

[6] The author obtained his NASD Commodity Futures (Series 3) trading license in 2003. He is trilingual and well-traveled. He has visited the BRICS at least once, except for Brazil and has lived and worked in four different developed countries. He has been teaching economics and business courses to undergraduate and graduate students since 2002 (e.g., principles of macro- and micro economics, strategic management, international marketing, global strategy) in the US and in Europe. He has published his expertise in several academic journals, as well as in the specialized press (e.g., *Les Echos* and the *Financial Times*). He is the recipient of several corporate, teaching, and service awards. Whereas his international business background is quite extensive, it must be noted that the researcher has natural biases.

[7] https://www.merriam-webster.com/dictionary/conventional%20wisdom.

**Assumption 2.** *Developed countries will always lead the global economy, others should follow their best practices.*

Since the end of WWII, industrialized nations have developed a new industrial order (Drucker 2017) and lowered trade barriers in order to boost global trade and foster peace among nations. This proven model for more than 70 years has propelled the US multinationals to unrivaled dominance. Therefore, all other countries tend to imitate the US (industrialized) model, especially developing or emerging countries.

**Assumption 3.** *Developed countries have better institutions. Therefore, people who lead them know better.*

Among modern institutions, universities are considered as some of the oldest ones. They date back as far as the Middle Ages and were established in Europe (Altbach 1998) and used Latin as the common language (e.g., Bologna University, est. 1088 in Italy) to create and disseminate knowledge. This century-old process has attracted millions of students from the BRICS to Western Europe or to other developed countries that, over time, replaced Latin with English, as the lingua franca among scholars. Nonetheless, it took centuries for universities to create the concept of a business school.

The world's first collegiate business school was created in Philadelphia, Pennsylvania in 1881 with a gift from entrepreneur and industrialist Joseph Wharton to produce graduates who would become "pillars of the state, whether in private or in public life."[8] Based on this intellectual legacy in industrialized or developed countries, the assumption has been that people who lead institutions in these countries know better.

**Assumption 4.** *Emerging markets cannot produce world-class companies.*

For decades, the concentration of power in US firms has been accumulated in systematic and strategic fashion (White 2002). This dominance has been the source of much admiration around the world thanks to the Forbes and Fortune 500 lists that have now become a ranking mechanism of reputation (Bermiss et al. 2013) and prestige. Most of the firms in those list are American companies. Therefore, it is easy to assume that emerging markets cannot produce companies that could come close to rival with a world-class American company.

**Assumption 5.** *Among institutions, universities and their business schools do not play a major role in society since business has been criticizing and ignoring business schools for decades.*

In the 1950s and as a result of in-depth reports (Gordon and Howell 1959; Pierson 1959), the American business schools' community was described as resembling trade schools which lacked academic rigor. These reports also recommended that business education leverages the model of medical and law school for business to become a profession so that business can find its place in society. Whereas the latter recommendation fell on deaf ears, the first one led to an (over)emphasis on analytical skills and theories. This produced more criticisms about business education lacking relevance (Pfeffer and Fong 2002) and the speculation as to whether the MBA will still exist in 2020 (Schlegelmilch and Howard 2011). In any case, the disconnect between business schools and the

---

8    https://www.wharton.upenn.edu/about-wharton/.

business community has been clear and widely publicized (Skapinker 2011, 2008). Based on this, it can be assumed that business schools do not play a major role in society.

As reasonable as these assumptions may appear to be, they need to be checked against the chronology of multiple events and patterns across countries, institutions, and industries.

**(B) Chronology: from the dream of the BRICS to the nightmare of the GFC**

In 2001 and in the developed world, many dreamed about the BRICs, not suspecting that a few years later, the 2008 GFC would become their worst nightmare. Trillions of dollars and millions of jobs lost, due to greed and miscalculated risks in the developed world gave way to the BRICS and their emerging multinationals to rise. Unbeknownst to most in the developed world, the GFC also created some deep cracks in society (increased income inequality, financial bankruptcies), whereas the BRICS started collaborating with one another like never before. A closer look at the chronology of these events and the trends that materialized informs us about the "big picture" that has affected the global economy by taking into account what happened in both developed and emerging markets with a special look at the automotive industry, among other industries between 2000 and 2018 and as follows:

- 2000: The MSCI (Emerging Markets) index doubles within two years to reach 500 for the time. The shift from developed to emerging markets is starting. After two drops to 230 and 250 in 2001 and 2002, respectively, this closely-watched index will climb above 1300 in October 2007 without suffering any major setbacks.[9]
- 2000: Tata Tea buys Tetley Tea (UK).[10]
- 2000: John (Jack) C. Bogle, legendary investor and founder of the Vanguard Group ($5 trillion of assets under management, as of 2018), warns investors about the dangers of a "sound premise" and provides timeless advice against conventional wisdom (Bogle 2000).
- 2001: Jim O'Neill coins the term BRIC.
- 2001: China joins the WTO.
- 2002: UN Secretary General, K. Annan, introduces the idea of inclusive globalization in a speech at Yale University (Gemmill et al. 2002).
- 2005: Lenovo buys IBM's PC division.
- 2007: against conventional wisdom on subprime loans and housing prices, John Paulson, hedge fund manager, places his bet and executes the best trade ever, being rewarded with billions of dollars and fame (Zuckerman 2010).
- 2008: The GFC starts with the collapse of Lehman Brothers (Mensi et al. 2016). The largest bankruptcy in US history; i.e., more than $600 bn, went against the conventional wisdom that Lehman was too big to fail and was going to be rescued by the US government like other financial institutions. It wasn't and its collapse had many consequences in the financial markets, across industries and countries. It bankrupted two out of three of the American BIG THREE. It left Ford moribund with record losses of $14.6 billion and fueled the growth of EMNCs, such as Tata Motors and Geely. Except for Russia, which goes into isolation, the BRICS' financial markets recouple with that of the US with increased linkages (Mensi et al. 2016) and later demonstrate the importance of stocks from developed countries in optimal portfolios (Mensi et al. 2017). High risks typically call for high rewards that will be delivered by the US stock market in a subsequent long-term bull market.

---

[9]    https://seekingalpha.com/article/211305-the-shift-from-developed-to-emerging-markets-what-does-it-mean-for-investors.

[10]    https://www.indiatoday.in/magazine/cover-story/story/20091228-2000-tata-tea-tetley-merger-the-cup-that-cheered-741660-2009-12-25.

- 2008: Tata Motors buys Jaguar-Land Rover from Ford for $2.3 bn in cash.[11]
- 2008: China becomes a vital market for Ford since it passed the US as the largest market in the world for all automakers in 2010 and became their largest foreign market.
- 2008: According to McKinsey, the number of connected devices equals the world's population.[12]
- 2008: Lenovo becomes the sponsor of the Beijing Olympics.
- 2009: Bernie Madoff is convicted and sentenced to 150 years in prison for the largest Ponzi scheme in history.
- 2009: GM files for bankruptcy after more than 100 years in business.
- 2009: from an all-time high of 1337 in 2007, the MSCI (Emerging Markets) index plummets to 499 in February 2009.[13]
- 2010: Geely acquires Volvo cars from Ford and could have been seen as a partner. However, additional moves by the EMNC indicate that it is becoming a competitor to reckon with, both in China and at home, not to mention in Europe.
- 2010: Grupo Bimbo (Mexico) acquires the North American fresh bakery business of Sara Lee.
- 2011: MSCI (Emerging Markets) index recovers from the 2009 crash and reaches 1204 (see footnote 13).
- 2012: the MOOC revolution starts: greater access to business education at a fraction of the cost. MOOC courses are offered to millions by Wharton, Harvard, MIT, etc.
- 2012: The end of globalization? (Rugman 2012). The conventional wisdom about the benefits of globalization starts to change. Globalization becomes the enemy of many. Populism (Mudambi 2018) and protectionism grow in the UK, US, Spain, Italy, Russia, China, and Brazil.
- 2012: Dalian Wanda Group (China) acquires AMC Theaters for $2.6 bn.
- 2013: Jim O'Neill retires from Goldman Sachs as Chairman of the Asset Management Division.
- 2015: Goldman Sachs closes its BRIC fund after years of losses and assets under management free-falling from $800 to $100 million.
- 2015: LinkedIn reaches 500 million users and is increasingly used by scholars to socialize their research.
- 2016: Brexit vote takes place. It catches financial markets and political science experts by surprise. Unaware of the consequences of leaving Europe, millions of British turn to google to better understand the EU that they just rejected in a historic vote.
- 2016: Haier buys GE's appliances division for $5.4 billion.
- 2016: Donald Trump is elected President of the US.
- 2016: the MSCI (Emerging Markets) index plummets again to 742 (see footnote 13).
- 2017: President Trump announces his America First policy and exits the Transpacific Partnership Agreement which was going to reduce tariffs, create jobs, and boost trade.
- 2017: ChemChina completes the largest foreign takeover by a Chinese firm by acquiring Syngenta for $43 billion.[14]
- 2017: Geely buys Lotus (UK).
- 2018: the MSCI (Emerging Markets) index climbs to 1254 in January and loses 23.84% of its value by 2 January 2019, trading at the level of 955 (see footnote 13).
- 2018: amidst a trade war between the US and China ((Trump administration decision to impose 25 per cent tariffs on steel and 10 per cent aluminum imported from China, retaliation from China,

---

11  http://shares.telegraph.co.uk/news/article.php?id=2799469&archive=1&epic=500570.
12  https://www.slideshare.net/AhmedALBilal/2017-q2-mckinsey-quarterly-global-forces.
13  https://www.investing.com/indices/msci-emerging-markets-historical-data.
14  https://www.reuters.com/article/us-syngenta-ag-m-a-chemchina/chemchina-clinches-landmark-43-billion-takeover-of-syngenta-idUSKBN1810CU.

and escalation with other tariffs between the two countries), the conventional wisdom on the permanence of free trade agreements and free-trade policies started after WWII is rocked to its core. Major financial markets enter correction territory.

- 2018: Geely buys a stake in the truckmaker AB Volvo and took a 10% stake in Daimler (parent company of Mercedes-Benz). All of these brands are sold in the US.
- Late August 2018: Geely announces the building of a new plant to produce bigger vehicles (trucks, SUVs, and cross-overs) and expected 90% of its cars to be electric by 2020. This confirms the growth of Geely both as an automaker but also as a threatening competitor to Ford.

If we compare and contrast the assumptions with the events and patterns above, we might be able to uncover some unconventional wisdom, also known as strategic insights.

## 3. Results

*Finding Some Unconventional Wisdom to Derive Strategic Insights*

In this section, we aim to find new strategic insights by reviewing the five assumptions based on the above chronology.

**Assumption 1.** *In an increasingly protectionist world, a domestic mindset is the way to go.*

Based on decades of research, the IMF re-affirmed in 2001, the year that China joined the WTO, that:

> "Policies that make an economy open to trade and investment with the rest of the world are needed for sustained economic growth. The evidence on this is clear. No country in recent decades has achieved economic success, in terms of substantial increases in living standards for its people, without being open to the rest of the world."[15]

Between 2001 and 2016, even with globalization suffering an identity crisis, the world remained more global-minded than protectionist and a global mindset was proven to be a predictor of success in global leadership positions (Javidan and Teagarden 2011; Beechler and Javidan 2007; Lane et al. 2009).

A research update as to whether tariffs are protecting American jobs also confirms that this is not the case since Ford prepared mass layoffs at the end of 2018 in order to off-set some $1 billion lost due to tariffs imposed by China in retaliation to US tariffs.[16] Other industries and more than 200 companies were also suffering from the tariffs in the US, cutting costs through layoffs and fore-going expansion.[17]

With the Smoot-Haley Tariff Act of 1930 in the US (Kennedy 2003) and the Imports Duties Act of 1932 in the UK, history has also shown that protectionism can be destructive rather than protective. In 2010, Oxford University Press published a great analysis which explained that "Hoover refused to listen to the pleas of 1,038 American economists who, in 1930, urged him to veto the Smoot–Hawley tariff bill. When it became law, this legislation raised US import duties and ultimately led to retaliatory action throughout the world. Not surprisingly, US foreign trade declined once the depression began to bite" (Crafts and Fearon 2010).

Clearly, this assumption is invalidated. It is a myth based on fear (Kennedy 2003). By definition, the 21st global economy cannot function based on a protectionist mindset. Globalization may be changing. However, the world remains inter-dependent and semi-globalized, where global trade represents at least 30% of the global economy (Ghemawat 2018).

**Assumption 2.** *Developed countries will always lead the global economy, others should follow their best practices.*

---

15   https://www.imf.org/external/np/exr/ib/2001/110801.htm.
16   http://fortune.com/2018/10/09/ford-stock-today-layoffs-trump-trade-tariffs/.
17   https://www.cato.org/publications/commentary/here-are-202-companies-hurt-trumps-tariffs.

This assumption is based on the current context, a view that suffers from myopia. When taking a much broader look at business and economic history, we should remember that China and India were dominant countries during previous centuries and that Russia was a mighty empire led by strategic tsars.

By focusing on China alone, it must be noted that the Han Dynasty established as a major power more than 2000 years ago.[18]

In all fairness, China's current economic development, growing faster than developed nations (e.g., Japan) for decades since the 1980s (Kang 2007) but still considered "developing" due to institutional voids (Khanna and Palepu 2010, 2005), cannot be the only period used to assess this 5000 year old civilization that goes back much farther in history than most developed countries. For example, it must be noted that China developed fast when Europe was experiencing turmoil during the Middle-Ages after the fall of the Roman Empire. According to a highly-cited book, published by the OECD, an important development was the "Chinese settlement of the relatively empty and swampy lands south of the Yangtse, and introduction of new quick–ripening strains of rice from Vietnam suitable for multicropping" between the eighth and thirteenth centuries (Maddison 2007, p. 18). During these five centuries, "population growth accelerated, per capita income rose by a third, and the distribution of population and economic activity were transformed. In the eighth century only a quarter of the Chinese population lived south of the Yangtse; in the thirteenth, more than three quarters. The new technology involved higher labour inputs, so productivity rose less than per capita income" (Maddison 2007). Today, China and other countries are considered developing countries or emerging economies whereas they once dominated, similarly to the Roman Empire. Dominance is not permanent. No developed country can aspire to always lead the global economy. Last but not least, adopting policies from the West or the "developed" nations, also referred as the "Washington Consensus," with the premise that they would be successful elsewhere, can result in economic disaster in developing nations (Chang 2002). Therefore, this assumption is also invalidated.

**Assumption 3.** *Developed countries have better institutions. Therefore, people who lead them know better.*

Institutions, such as universities and business schools, can create and hire scholars who can come from any country in the world and many have come from the BRICS. Scholars can also be born out of institutions, even if it remains an exception rather than the norm. Indeed, one of the foremost mathematicians who was born in India, Srinivasa Ramanujan, solved very complex problems and advanced theories through a series of breakthroughs in his youth without having much formal education (Singh 2017). Other scholars from Russia include many Nobel Laureates in several disciplines, especially physics (Dardo 2004). For all these reasons, the assumption on the "superiority" of institutions in developed countries and of the knowledge accumulated by the people who lead them is invalidated.

**Assumption 4.** *Emerging markets cannot produce world-class companies.*

This assumption is also based on the perception that emerging markets are "not there yet" (Khanna and Palepu 2000; Khanna and Palepu 2010) and therefore cannot compete with industrialized countries' giants, such as IBM, Boeing, Ford, Apple, etc. The reality is much different: Lenovo, once a regional player has become the largest PC vendor in the world after acquiring the PC division of IBM. Embraer (Brazil) is another success story, and so is Tata Motors which acquired Jaguar and Land Rover from Ford in 2008, or even Huawei overtaking Apple in global smartphones sales in 2018.[19] These are just a few examples among many. A closer look at the largest companies in the world, measured by

---

[18]   https://www.bbc.com/news/world-asia-pacific-13017882.
[19]   https://qz.com/1345496/apple-was-just-overtaken-by-huawei-in-global-smartphone-sales/.

revenues, also indicates that 3 out of the first 5 largest are Chinese.[20] This assumption is a myth that is easily busted once researched.

**Assumption 5.** *Among institutions, universities and their business schools do not play a major role in society since business has been criticizing and ignoring business schools for decades.*

This assumption is often based on the misunderstanding associated with the value and contributions of education to society. According to a 2017 UNESCO report, the total number of higher-education students has doubled between 2000 and 2014[21] and represents a healthy 5% increase on an annual basis. Business education is the most popular choice among all disciplines; it attracts about a quarter of all university students. This assumption is often based on a quantitative analysis which does not account for qualitative contributions, such as launching new businesses, finding solutions to societal problems, etc. Business education has even transformed an emerging market like China by going from Marx to a hybrid capitalistic model (Tsinghua)[22] or profoundly boosting the country's competitiveness: India (Indian Institutes of Management), FGV (Brazil, and Skolkovo (Russia). Therefore, this assumption is also invalidated.

Many of the higher education students earn an MBA, the passport to management and especially to careers in Finance or Consulting. Business actually touches all disciplines and vice versa (e.g., medicine). To better assess the role of business schools in society, we must wonder how the world economy would perform without them and without business education in general. Clearly, if the BRICS have made so much economic progress, a large part of that can be attributed to education, the social elevator, and one of the best ways to help a country navigate the global economy competitively. Education provides the tools to make better decisions, solve problems, and redistribute wealth. In both emerging and developed countries, what is needed is not less education but more. We will discuss this in greater details below.

## 4. Discussion

Invalidating assumptions that seemed reasonable at first can lead to a state of confusion. What should we believe now? This is where higher-education and business schools, in particular, have a particularly important role to play and can make a positive and global impact on society. Business schools teach the hard skills (e.g., research, analytics, problem-solving, and reasoning) and the soft skills (communication, cross-cultural literacy, and conflict resolution). In addition, business schools socialize their students with the rest of the university and the business community through general education courses on ethics or other matters, such as political science, computer science, and foreign languages. Students, not only acquire knowledge that is crucially important in the 21st century economy that is more internationally interdependent than ever, but also learn appropriate behaviors and perspectives unlike their own. These skills and behaviors applied to the problems faced by today's society can advance all of us toward a resolution that is acceptable to all (e.g., sharing values and reducing income inequality). In short, business education can help us coexist and reduce conflicts within each country and between countries. To achieve this collaborative approach and avoid divisive issues, an inter-disciplinary education is very important. However, all too often, students focus on one discipline whereas problems call on several disciplines to be solved. These and our society are more complex than ever. Therefore, we must get both the small and the big pictures right.

By combining the reasonable assumptions above which were based on a narrow perspective with a bigger picture approach, this paper aims to provide a better understanding of key issues that have been plaguing emerging markets. That big picture approach reminds us that the largest two of them

---

[20] http://fortune.com/global500/.
[21] http://unesdoc.unesco.org/images/0024/002478/247862E.pdf.
[22] https://money.cnn.com/magazines/fortune/fortune_archive/2005/05/16/8260218/index.htm.

by population (China and India) represent 40% of the world's population. We also need to think in human terms, not just in numbers terms. Qualitative analysis must supplement quantitative analysis to better understand progress to date, as well as the risks and opportunities that lie ahead.

For example, the pattern of world-class companies coming out of emerging markets has surprised many executives and investors in developed markets. In some cases, some of them misunderstood their competition and the threat associated with those emerging multinationals. Some scholars spotted the trend early on (Khanna and Palepu 2010; Guillén and García-Canal 2009; Ramamurti 2012; Holtbrügge and Kreppel 2012). However, due to the focus of most business executives on the small picture and short-term results, many of them did not benefit from these important research contributions. They were blindsided by the risks. It is, therefore, necessary to conceptualize a new and holistic framework [Figure 1] to challenge conventional wisdom in order to improve our assessment of risks and opportunities. This framework combines the quantitative and qualitative analyses of macro trends, micro behaviors (of institutions, firms, customers, especially if they become movements, such as populism), implications, critical thinking, and research. It should result in strategic insights that we call unconventional wisdom, as it challenges pre-conceived ideas and contributes to new conventional wisdom through a feedback loop of continuous improvement.

Education is a choice. We can choose to identify problems, prioritize them, and look for solutions that are acceptable to all and will avoid the worst catastrophic risks from materializing. An inclusive globalization based on shared values (Porter and Kramer 2011) would be advisable since we are all inter-dependent and "all in it together." To put this important concept in practice, business schools play a vital role in the economy and society, in regards to, not only create prosperity but also help re-distribute it, while staying relevant. They must train their students about that role and increasingly about disruptions that affect countries, people –their future and the future of work in the context of artificial intelligence (AI) advances, unparalleled until now—institutions, industries, and markets. Thankfully, business schools are not alone in this daunting tasks. Organized among peers by the Association for the Advancement of Colleges and Schools of Business (AACSB) International since 1916, world's largest business education alliance, they clearly benefit from valuable resources.[23] The new AACSB Chair as of 31 January 2018 shared the organization's new vision and the key success factors for business schools: (1) cultivate a position at the intersection of academe and practice, (2) be a driver of innovation in higher education, and (3) connect with other disciplines (Beck-Dudley 2018). Disruptions and change in general, provide challenges and opportunities. Business schools, as well as the graduates and scholars that they produce, have the unique capacity to address challenges and capitalize on opportunities. It will fascinating to watch how they evolve in society and what positive contributions they will continue to make.

## 5. Conclusions

Conventional wisdom, used knowingly or unknowingly, has not always been the friend of the strategist, especially in an emerging market environment where institutional voids, rampant corruption, and other dangers loom large and can be found next to unparalleled opportunities. Due to the scarcity of strategic frameworks to assess the BRICS since the GFC, this paper aims to help fill this gap by providing a simple but holistic and analytical framework that we call the Holistic Risk and Opportunity Analytical Framework (HROAF). It relies on both qualitative and quantitate data, as well as macro (trends) and micro (behaviors) analyses. It can provide the strategist with a different lens to manage a new normal of uncertainty, volatility, risks, and opportunities, thereby creating some unconventional wisdom and a sustainable competitive advantage for him or her. Whereas the HROAF should allow strategists to generate some initial insights, each strategist is expected to dive deeper in each portion of our multi-dimensional framework, as needed. We believe that, when used regularly,

---

23   https://www.aacsb.edu/newsroom/2018/2/new-leadership-named-to-aacsb-board-of-directors.

this framework could become a useful tool to unleash the potential of a new strategic and global mindset that will help reduce the biases and dangers that come from conventional wisdom.

Limitations: as useful as this approach may be, it has limitations. It is based on only one researcher, his experience, his research, and his natural biases. Only five assumptions were tested as a basis for illustration. It is possible that additional assumptions would unleash more insights. In addition, mixing methods can be challenging and require a large team supporting each strategist. Not everyone can expect to have such support to implement this framework.

More research, a deeper dive, and more triangulation are needed, especially on how deeply-rooted in conventional wisdom any assumptions are. In an era of fake news, scholars have the societal responsibility to redirect assumptions and bust myths that have been misleading investors, policymakers, and voters. Uninformed decisions can have catastrophic economic and human consequences for any given country and the world economy.

Using our Holistic Risk and Opportunity Analytical Framework to discover some much-needed unconventional wisdom and setting inclusive globalization as a priority could be a way to rebuild the BRICS and allow everyone to prosper in peace.

**Funding:** This research received no external funding.

**Conflicts of Interest:** The author declares no conflict of interest.

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
