# Peer review of "Using Unconventional Wisdom to Re-Assess and Rebuild the BRICS"

_jrfm, doi:10.3390/jrfm12010008_

Round 1

Reviewer 1 Report

This paper provides an interesting analysis on BRICS after the global financial crisis. The paper could be improved by consideration of the following points:

1.       You could discuss some of the literature between developed markets and BRICS during and after the global financial crisis. Please consider papers such as Mensi et al (2016, 2017).

2.       In terms of the role of business education you could also discuss papers such as Beck-Dudley (2018).

References

Beck-Dudley, C. (2018), The Future of Work, Business Education, and the Role of AACSB, Journal of Legal Studies Education, 35, 165-170.

Mensi W., Hammoudeh S., Nguyen D.K., Kang S.H. (2016), Global financial crisis and spillover effects among the U.S. and BRICS stock markets, International Review of Economics and Finance, 42, pp. 257-276.

Mensi W., Hammoudeh S., Kang S.H. (2017), Dynamic linkages between developed and BRICS stock markets: Portfolio risk analysis, Finance Research Letters, 21, pp. 26-33.

Author Response

Dear Reviewer 1:

Thank you for providing some useful and timely feedback. I agree and have made the following changes, accordingly:

- Ran grammar and spell checks to correct a few mistakes;

- Added Mensi et al 2016 and 2017 to the paper (Lines 141-148)

- Added Beck-Dudley (2018) and discussed the implications of the new AACSB vision (Lines 310-325).

I trust that these changes are satisfactory. In addition, I would like to also let you know that I added the framework and figure that conceptualize the approach discussed in this paper, as requested by the Managing Editor (Lines 295-301 and 304-305).

Respectfully,

The Author

Reviewer 2 Report

The paper attempts to point out some major misconceptions about the abortive development of the BRIC countries and potential re-building of the BRICS in the future.  Based on the conventional wisdom, the author delivers five assumptions attributable to possible cause of the BRIC's failure. In conclusion, the author argues that innovation can arise in any company and any country.  Economic perspectives from developed and developing countries are equally likely which refuting the conventional wisdom profoundly embedded in the five assumptions.

General comments:

In addition to qualitative argument, the author should provide more quantitative data to support the arguments of each assumption.

The discussion presented in Section 2 (Materials and Methods) appear to be redundant and unconvincing.

The author should relate some theoretical concepts to prevailing situations concerning the false assumptions.

Please find more detailed comments in the attachment.

Author Response

Dear Reviewer 2: 

Thank you for your feedback. Further to your comments, the following enhancements were made in the manuscript (abstract, introduction, methodology, results, discussion, and conclusion sections) in order to make it more convincing, in particular here:

- L 11-21: added quantitative contributions from HBS scholars on strategy in 12 emerging markets: Khanna and Rivkin (2001), as well as qualitative studies in top journals, such as JMS and JIBS: Wright et. al. (2005) "Strategy Research in Emerging Economies: Challenging the Conventional Wisdom [1,771 citations on Google Scholar) and London & Hart (2004) on reinventing strategies for emerging markets.

- L 49-50 & 157-159: included the landmark achievement; i.e., "best trade ever" of John Paulson who made billions by betting against conventional wisdom on the subprime loans and housing prices in 2007 (Zuckerman 2010). 

- L. 149-150: added Jack C. Bogle's timeless advice on the dangers of a sound premise (Bogle 2000).

- L. 161-164: included the Lehman Brothers' collapse against the conventional wisdom that expected this financial institution to be rescued by the US government.

- L. 187-190: Added the contribution of Rugman who called for the end of globalization and marks a change of attitudes (from positive to negative).

- L. 210-213: the conventional wisdom on the benefits of free-trade is starting to be challenged and morphs into new policies. A trade war follows.

- L. 363-400: rewrote conclusion to make the contribution clearer and explained the role of our Holistic Risks and Opportunities Analytical Framework in creating unconventional wisdom for each strategist to gain a sustainable competitive advantage.

Lastly, further to Einstein's position on measurements, we feel that "not everything that can be counted counts and not everything that counts can be counted" (n.d.). The beauty is in diversity.

In conclusion, we trust that these enhancements meet your expectations and look forward to hearing from you.

Respectfully,

The Author

Reviewer 3 Report

This is a well written piece of academic work. I consider that it should be published after some minor adjustments.

(i) I think that bio text (lines 51-59) should be presented either as a footnote or as a specific section at the end of the manuscript. Subsequently, the paragraph starting from line 60 should be slightly amended.

(ii) sometimes an assumption could be invalidated via a counter-example. According to many sources, Huawei overtaken Apple on mobile phones market. Hence, the sentence from line 106 could be adjusted. Later edit: I accomplish that later on the assumptions are invalidated (lines 215-264). May be Huawei example could be added to paragraph which starts on line 239 and ends on line 247.

Author Response

Dear Reviewer 2:

Thank you very much for your timely, useful, and positive feedback. I have made the following changes accordingly:

(i) Lines 51-59: bio placed in the footnote section. Line 60: subsequent text was amended slightly.

(ii) Lines 246-250 now include the excellent example that you submitted regarding Huawei overtaking Apple in global smartphone sales.

In addition, I corrected a few grammatical and spelling errors. I trust that these changes are satisfactory.

Lastly, I would like to also let you know that to further explain the method, I added the framework and figure that conceptualize the approach discussed in this paper. Please see Lines 302-307 and 309-310. This is consistent with another request from the Managing Editor.

Respectfully,

The Author

Round 2

Reviewer 2 Report

The revised version of the paper is not much improved from the first draft.  In the introduction, it is unclear what the findings of Khanna and Rivkin’s (2001) quantitative study imply any inadequacy of conventional wisdom on emerging markets.  The author also needs to be more specific about what the strategy research recommendations from Wright et al (2005) and London & Hart (2004) are on the challenge of the conventional wisdom on emerging economies. 

The validity of each of the five assumptions is still not clear. 

For rebutting assumption #1, the author needs to explain why trade openness is a better strategy than protectionism with sound theoretical reasoning. The argument refuting the assumption of protectionism is logical but not theoretical.  Evidence to affirm the rebuttals to assumptions #1 and #2 are also required.

Author Response

Dear Reviewer #2:

Thank you for your feedback. The requested improvements and clarifications were made. They are detailed in this 3-page document. I trust that they will meet your expectations.

Regards,

The Author
